

**The Flat Earth satire, using science theatre to debunk absurd theories.**

George Sand França[1], Ricardo Cruccioli Ribeiro[2], Luana Rosa Soares[3], João de Oliveira Soares[3], Gabriel B. de França[4],Paulo Eduardo Brito[5].

5    [1]Observatório Sismológico, Universidade de Brasília
[2]Faculdade de Educação, Universidade de Brasília
[3]Faculdade de Artes Dulcina de Moraes
[4]Instituto de Química, Universidade de Brasília
[5]Faculdade Planaltina, Universidade de Brasília

Correspondence to: George Sand França (georgesand@unb.br)

**Abstract**: Science needs everything, and art must feed on it for its actions. With the growing popularity of social media, absurd theories have been gaining
consensus without any criticism, and, even worse, they have consolidated. Thus, in order to find solutions for a better understanding of our theories, it was created the project "The Earth is Flat! Now What? ", that uses performing art as the main communicator to spread science. The first step was a meeting for promoting integration among Geophysicists, Art-educators, Artists, Astronomers, clowns and
all the different expertise participating in the project. The meeting was also the occasion for planning the show. The second step was the dramaturgy along with the creative process, which involved discussions of the scenes and lessons about the theme to be presented. And the third step was the performance. The bibliographical review, which is the same method named the "table's work" for
artists, was not based on indexed journals, but rather on social networks and classes for understanding the shape of the Earth. The show impacts the community in a fun way offering the opportunity of a new experience to the population.



**Introduction**

The recent growth of social media environments has made absurd themes and theories a considerable growth in society. Sciences have no actions or initiatives. There seems to be no concern with topics such as flat earth, creationism or global warming. Mostly, the scientific dissemination is limited to children, with no attention

for the adult audience, although we have examples like the series "The Big Bang Theory", there is still no strand coming from the academy, especially in Brazil, sometimes some actions, but with low popularity. Searching for this connection between science and art is certainly complicated, due to the exhaustive scientific activity of researchers, which most of the time there is no way to present

themselves in the art form, however we can unite with educating artists and assist or boost our art science. To this end, the theatrical process "The earth is flat! now what?" (in portuguese: A terra é plana! E agora?), where his first performance took place at the European Geoscience Union (EGU) 2019 General Assembly (França et al. 2019a) . That seeks this link through the clowning between science and art

and in this work the whole process of creation will be shown in a summarized form and also the results obtained from the audience through the google form for the public that attended.

**Art-Science and the process**

Palmas (2006) reports in his experience with science and art, that theater is a powerful tool for scientific dissemination. In his Art and Science project on stage, he reports on the experiences and difficulties in implementing this integration. The project won several awards and had a great repercussion in the community.

Although the dialogue between science and theater is old, it is still a localized movement with little material available, especially in Brazil. Almeida et al. (2018) presented their experience of integrating science and theater at the Science and Life museum and the museum of life. Almeida et al. (2018) has a successful and continued project that drives this integration. Our work enters the area of earth

science and art, where it is still very little explored and we use the same tools used





by previous works

The information was collected on this subject using the social networks that defended this false theory and through scientific dissemination that refute the theme, scenes for spectacle were simultaneously created. The show's introduction

scenes were selected, in which we used two "mamulengos" to present the show. After the presentation, a short lecture that will be disturbed and at least three slides. For the next scene, we break the text of the big bang theory, adapted from Oliveira 2018, in which we use of body as language. Given continuity, music inspired by the opening of the series The Big Bang Theory, by Thomazoni W.

(2013) was used and adapted for the show. In addition, scenes from França et al. 2019 (Figure 1) featured Newton's scene and gravity in the EGU session. Finally, the remaining three scenes were: the story of the planets with balloons and a Pilates ball; based on the film "the great dictator" by Charlie Chaplin; and finally the Earth is Flat sermon. These scenes were created for about six months, being

evaluated weekly. As for the costume, a black suit with red details was established trying to describe a speaker, initially a checkered pants, which were replaced by a brown shorts. Wine Socks, white shirt and suspenders to complete the costume. The dramaturgy was a compilation of the scenes described above, however the text was described after the presentation of the open rehearsal. The artists and

scientists from the city were invited to a conversation and discussion about the show before the  first run. With everything ready, leaving only the scenario we chose in a scenario that facilitated mobility, for this we placed a folding screen, a pulpit, a ball and a transport crate.

In total there were 7 shows in a theater, and two turns at international conferences

(França et al., 2019a, 2019b), considering only the audience of the seven performances we had a total of 316 people. The venues of the presentations were at FAP-DF on October 12th, at Casa dos Quatro from November 15th to 17th, at Oficina Perdiz, on December 21st to 22nd and participating in the FICA festival of Casa da Ribeira in the city of Natal -RN, on December 29th. The presentation

posters and photos are shown in Figure 2.

**Results**



To evaluate the processes, we used the audience that watched through a google form, that was made available. This form had 9 questions and an optional comment

Of the 316 people who have watched the show so far, 11% of the public have answered the questionnaire. A low number is probably due to the fact that the request only happened through several partners and after the last presentation. However, the questionnaire brought us several reflections both in the artistic and scientific dissemination.

The histogram of the audience and the responses shows in Figure 3 that about 40% of the responses were from the show held in Natal-RN, at Casa Ribeira, consequently the largest audience, followed by the presentations in Brasília, the season at Casa dos Quatro 37.1% (three shows) and 25.7% Teatro Oficina Perdiz (two shows). This division shows that we had a well distributed response and there was a mix of the audience from the academy and also an audience, since the presentation in Natal, had a more diverse audience since it was a theater festival - FICA, our process received the first criticism from the collective farofa critic. This review is available in Portuguese on the Farofa Critica's website (www.farofacritica.com.br/criticas/leiamais/137).

The majority of the public had a high degree of understanding of the shape of the Earth, 82.8%, certainly the title makes the public more aware of the subject Figure 3a. The number drops when the questioning is a little more specific, however it is well discussed in high school. The question was about the law of universal gravitation that 69.6% knew about the law of universal gravitation (Figure 3b). We finished this part of heliocentric and 20% answered that they did not know what it was (Figure 3c). This result brings us to believe that we have an audience with excellent knowledge and that apparently can lead us to greater understanding with laughter.

The second part is about on scientific dissemination, first question is one word was asked to represent the show, figure 4 shows a word cloud with emphasis on didactic, interesting, fun, playful, genius. This shows that we had a general acceptance of the public. If the show is science dissemination with a score of 1 to



5, 83% gave a score of 4 and 5 and 17% gave a score of 3, again it shows the importance of this play with a link in a space that is at least used in Federal Capital in Brazil (Figure 5).

As for the curiosity of the shape of the Earth, only 40% became even more curious, this is due to the fact that the vast majority are aware of the shape of the Earth and that certainly believe that another way is not feasible (Figure 6)

The show regarding classification and recommendation to another person obtained results similar to that of scientific dissemination, with another 80% with a score between 4 and 5. What we can conclude was that it was well accepted by the public (Figure 7).

As for the comments, the highlight is that the show is not ready yet, although all this result, especially the end of the show and why we use of the "mamulengos". That it was necessary in order to highlight the training of the actor in terms of qualification in science. The need to further highlight the issue of the flat Earth. Performance problem, p.e. physical preparation was also highlighted. With these comments, we are updating the play to seek the best show.

**Discussion and conclusion**

The long process of creating a show that was finally presented and showed that there is an excellent way to publicize it. The public that for the most part had prior knowledge on the subject, considered the character of scientific dissemination and also considered it an excellent show. The show had a good repercussion in society, winning the DF 2019 Theater Award Category: Local Circulation Show.

An importance in the dissemination of sciences through Art, not only opens a space for science, but also presents to an audience that is not frequent in any theater or theater. The possibilities for exploring this path are endless and still few. We need to link art with science and strengthen this area of education and art. When you get stronger and do an artistic work based on the experiment to further guarantee the quality of scientific dissemination.

Scientific dissemination should be one of the main functions for researchers, who normally participate as collaborators in most dissemination activities. When you





participate in these activities, there is no recognition or accounting for your academic careers. Knowing how to explain your activity or link with art can bring more popularity to science, as well as to knowledge, stimulating and boosting science in all spaces.

## Acknowledgements

We are thankful to Hugo Freitas and Lyanna Soares by the photos. GSF thanks CNPq for the PQ 423 grants.

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



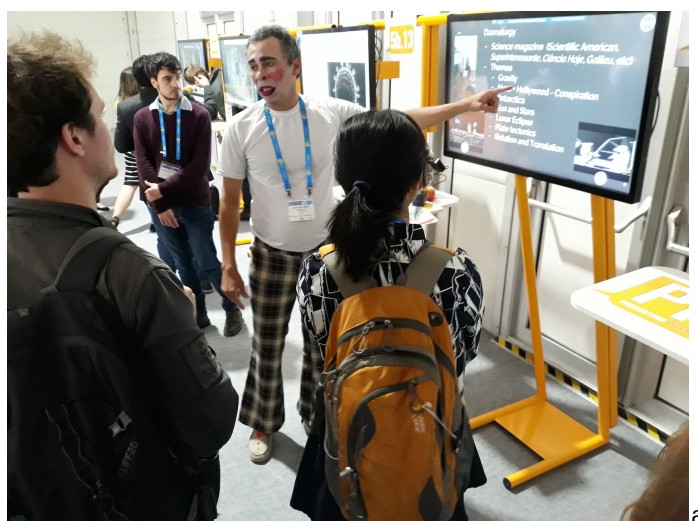

a)

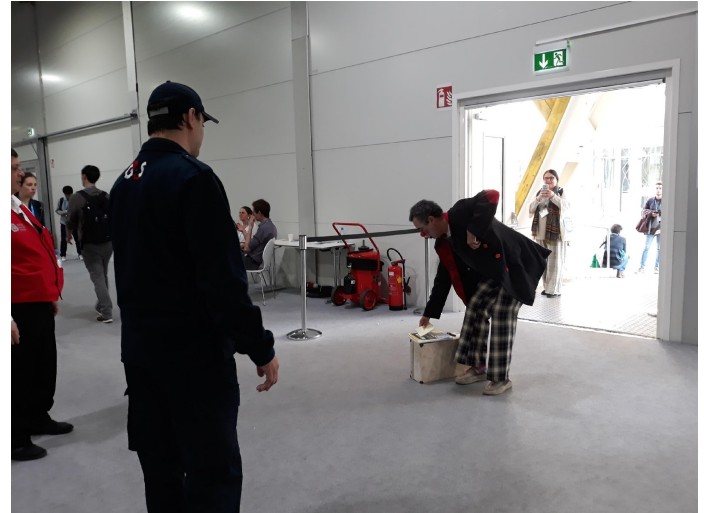

b)



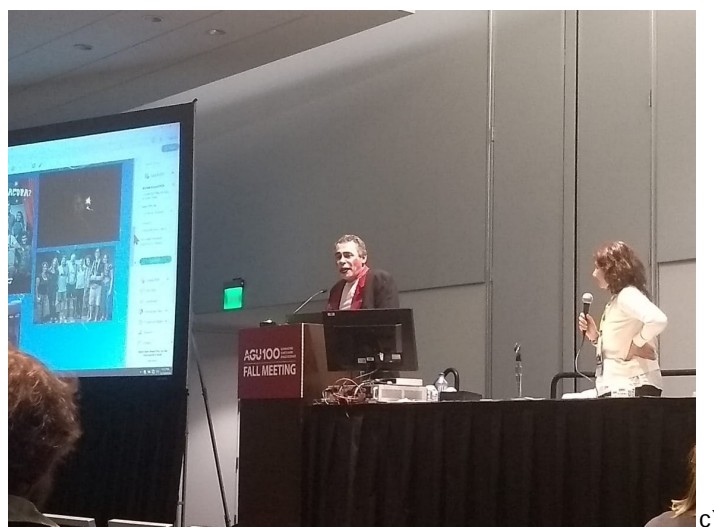
c)

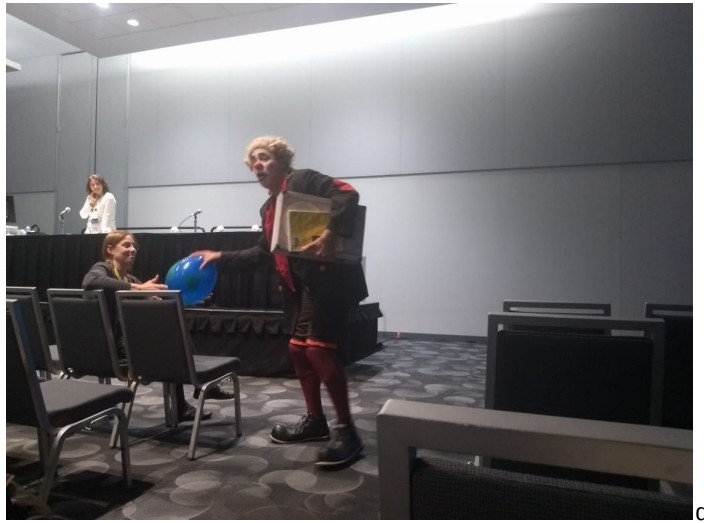
d)

Figure 1 :a) First presentation at EGU 2019, where it all started. b) Arrival at the

EGU 2019 congress in Vienna! c and  d) AGU Fall meeting 2019 in San Francisco!



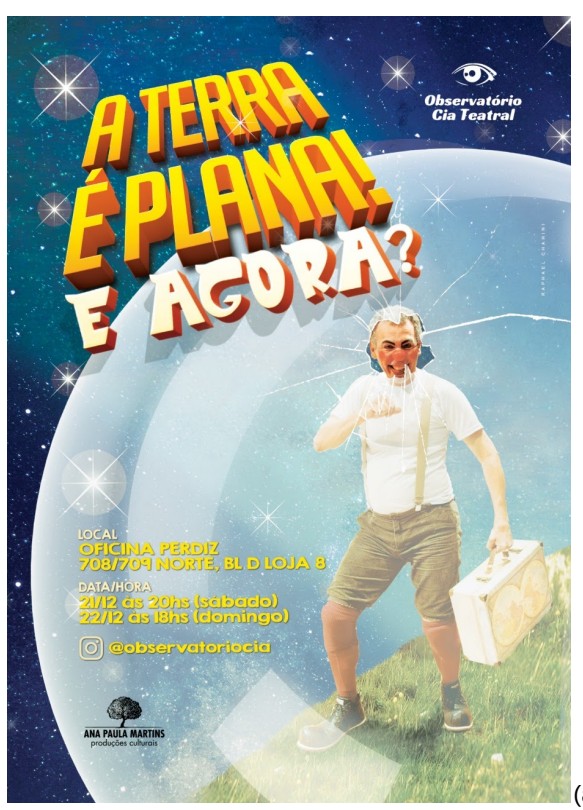

(a)

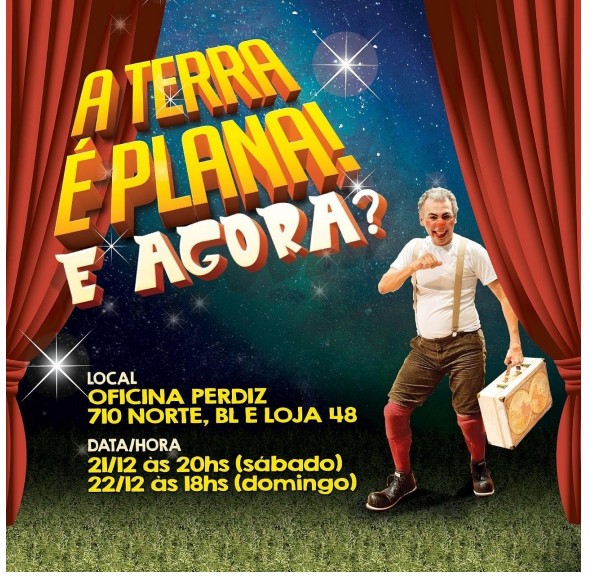

(b)



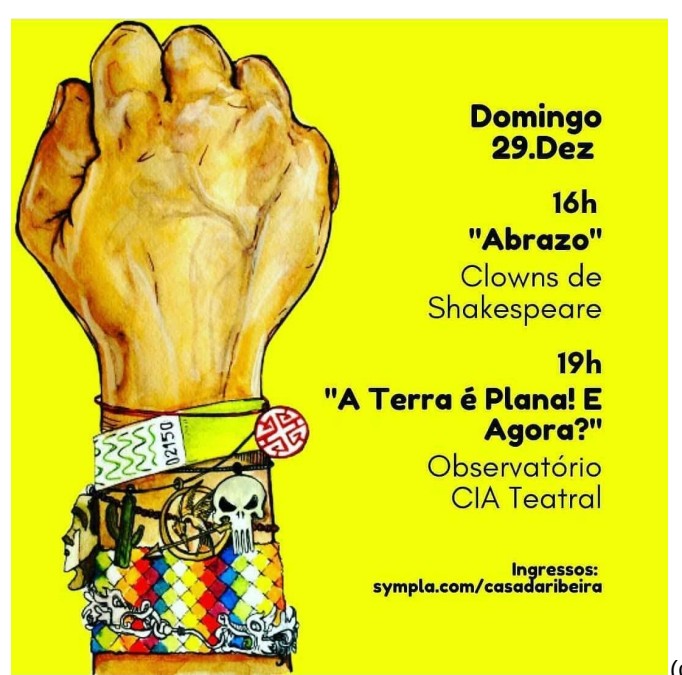

(c)

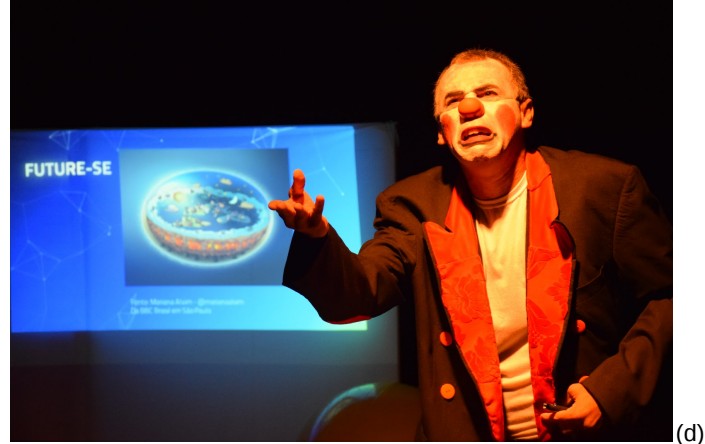

(d)



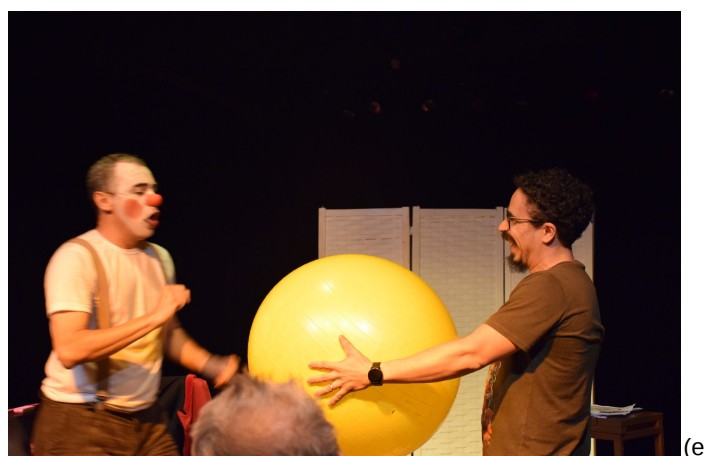

(e)

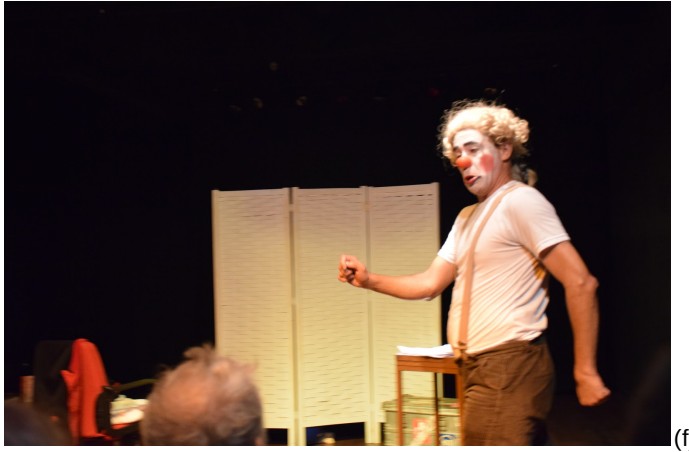

(f)

Figure 2:The porters of the presentations at Casa dos Quatro (a), Teatro Oficina Perdiz(b) and Casa da Ribeira (c) and photos(d,e,f) of the show (taken by Hugo de Freitas).



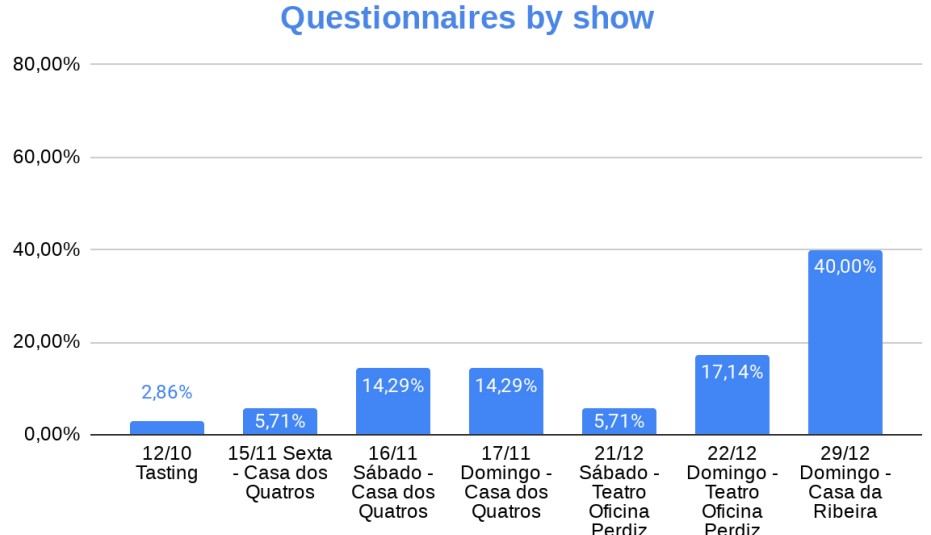

Figura 3: Histogram of audience and responses.

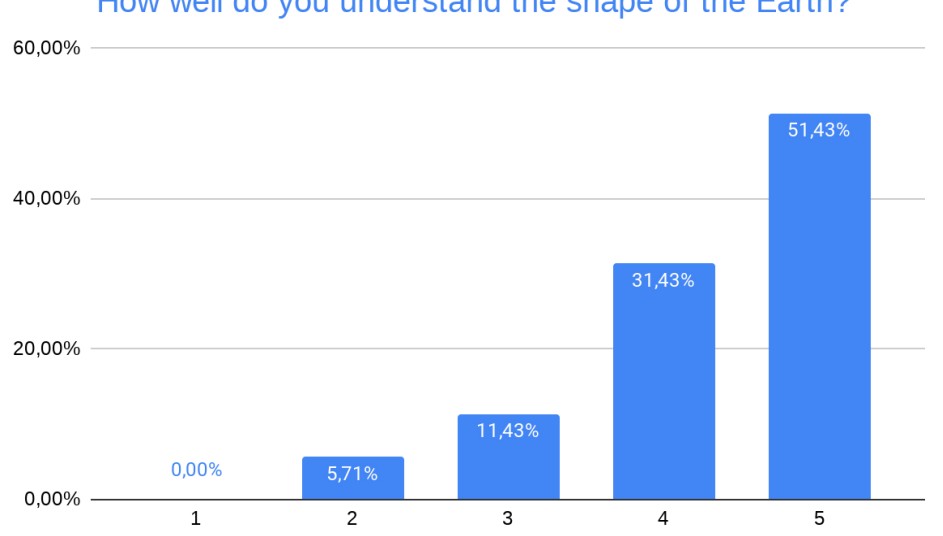





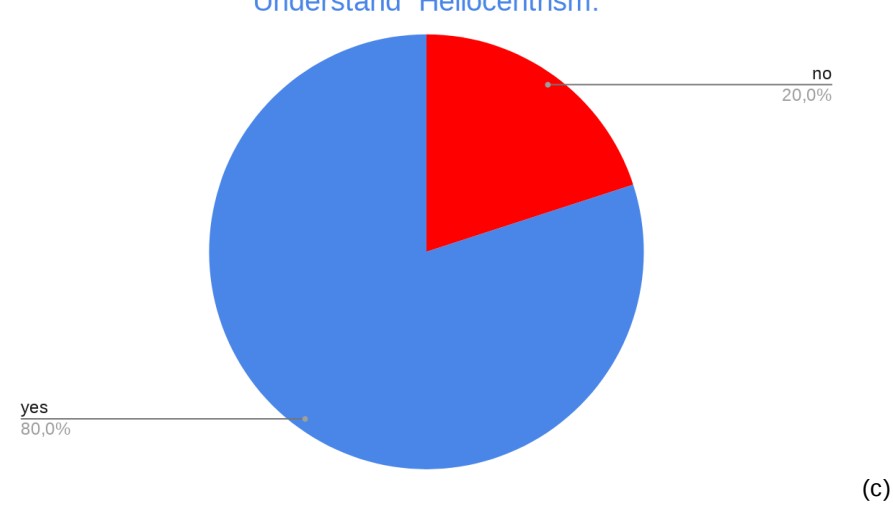

(a)

(b)

(c)

Figure 3: Graphics a) What do you understand about the shape of the earth b) What do you know about universal gravitation and c) What is heliocentric





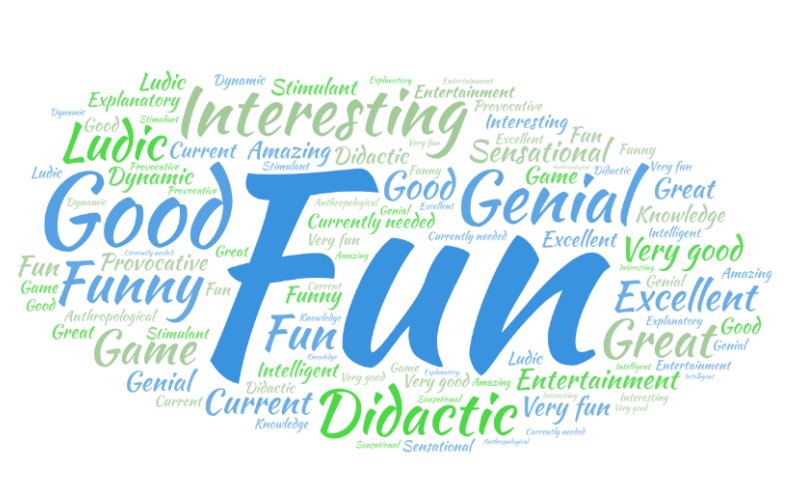

Figure 4: Word cloud shows the main words the presentation represents.

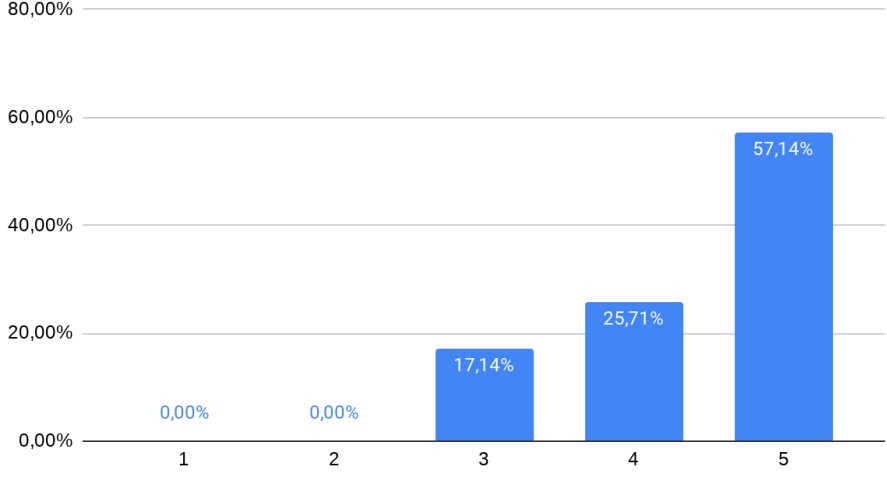

Figure 5: Graphic if the show promotes science.



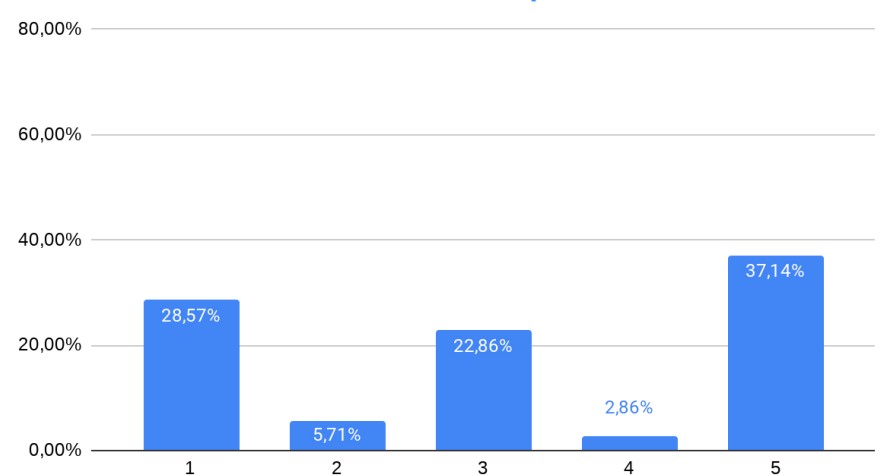

Figure 6: Graphics if more curious about the shape of the Earth.

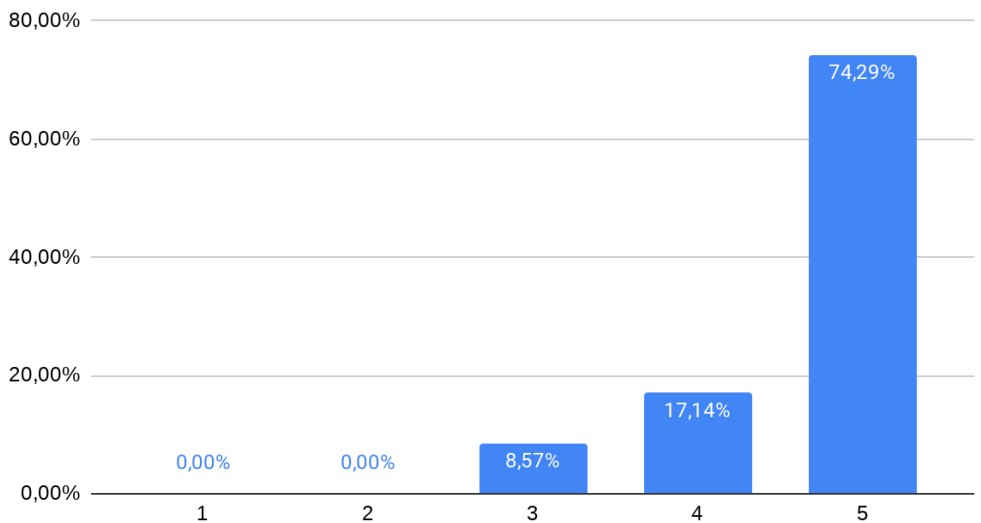

Figure 7: Graphic about the recommendations.