# Peer review of "The Flat Earth satire: using science theater to debunk absurd theories."

_Geoscience Communication, 2020_

## Referee Comment (RC1) · Juliana Garrido Damaceno (Referee) · 15 Jun 2020

The paper titled 'The Flat Earth satire, using science theatre to debunk absurd theories' presents relevant scientific information about a concrete initiative in communicating science by disseminating the knowledge by art performances to the scientific and general public. A questionnaire was presented as a way to evaluate the engagement of the public in the topics addressed. The paper in fact brings novel ideas in spread the scientific knowledge outside the academia, however in a ludic way, giving opportunity for whole families to think more about topics so essential in the development of scientific thinking. The authors gave appropriate credits to related works but I suggest a more deep investigation in the theme. The paper should return to the authors and pass by a better writing review. The authors are dealing with an important topic and should put

more effort into passing on the information in an appropriate way so that more good initiatives like this can emerge. If the paper return by the authors in the best written form, I recommend it to be published. Specific comments: The work won the DF 2019 Theater Award Category: Local Circulation Show and this information should be present in the introduction, the results and conclusion should be elaborated highlighting the fact that part of the public lack basic knowledge of scientific information, and not that great part of the public knows the shape of the Earth. This is important to highlight the need for initiatives in bring the science close to the general public. Please add more information about the public that filled the form (age, level of education, nationality, etc); I appreciated the conclusions but I also expected to know more about the next steps in developing this project.

• I suggest to remove the phrase 'Sciences have no actions or initiatives.'; • I suggest to rewrite more clearly the paragraph that starts in line 52: 'Searching for this connection between science and art is certainly complicated, due to the exhaustive scientific activity of researchers, which most of the time there is no way to present themselves in the art form, however we can unite with educating artists and assist or boost our art science.'; • I suggest to rewrite more clearly the paragraph that starts in line 59: ' . That seeks this link through the clowning between science and art and in this work the whole process of creation will be shown in a summarized form and also the results obtained from the audience through the google form for the public that attended.'; • I suggest to rewrite more clearly the paragraph that starts in line 74: 'Our work enters the area of earth science and art, where it is still very little explored and we use the same tools used by previous works (what tools?) The information was collected on this subject using the social networks that defended this false theory and through scientific dissemination that refute the theme, scenes for spectacle were simultaneously created.' • I suggest to rewrite more clearly the paragraph that starts in line 79: '. The show's introduction scenes were selected, in which we used two "mamulengos" to present the show.' Please could you explain what 'mamulengos' are? • I suggest to rewrite more clearly the paragraph that starts in line 81: 'After the presentation, a short lecture that will be disturbed and at least three slides. For the next scene, we break the text of the big bang theory, adapted from Oliveira 2018, in which we use of body as language. Given continuity, music inspired by the opening of the series The Big Bang Theory, by Thomazoni W. (2013) was used and adapted for the show. In addition, scenes from França et al. 2019 (Figure 1) featured Newton's scene and gravity (please we need more explanation about) in the EGU session. Finally, the remaining three (or two?) scenes were: the story of the planets with balloons and a Pilates ball; based on the film "the great dictator" by Charlie Chaplin; and finally the Earth is Flat sermon.' • I suggest to rewrite more clearly the paragraph that starts in line 96: 'With everything ready, leaving only the scenario we chose in a scenario that facilitated mobility' • The reference lines disappeared in page 4 • I suggest to rewrite more clearly: 'we used the audience that watched through a google form, that was made available. This form had 9 questions and an optional comment Of the 316 people who have watched the show so far, 11% of the public have answered the questionnaire' • I suggest to rewrite more clearly: '. This division shows that we had a well distributed response and there was a mix of the audience from the academy and also an audience, since the presentation in Natal' • Add parentheses (Figure 3a) in: 'certainly the title makes the public more aware of the subject Figure 3a.' • I suggest to rewrite more clearly: 'This result brings us to believe that we have an audience with excellent knowledge and that apparently can lead us to greater understanding with laughter.' • What is the purpose of the play since the audience already has the knowledge of the topic? • Authors should be more careful with punctuation and double spacing; • I suggest to rewrite more clearly: 'The second part is about on scientific dissemination, first question is one word was asked to represent the show, figure 4 shows a word cloud with emphasis on didactic, interesting, fun, playful, genius. This shows that we had a general acceptance of the public. If the show is science dissemination with a score of 1 to 5, 83% gave a score of 4 and 5 and 17% gave a score of 3, again it shows the importance of this play with a link in a space that is at least used in Federal Capital in Brazil' • I suggest to rewrite more clearly: 'The show regarding classification and recommendation to another person obtained results similar to that of scientific dissemination, with another 80% with a score between 4 and 5. What we can conclude was that it was well accepted by the public (Figure 7).' • I suggest to rewrite more clearly: 'As for the comments, the highlight is that the show is not ready yet, although all this result, especially the end of the show and why we use of the "mamulengos". That it was necessary in order to highlight the training of the actor in terms of qualification in science. The need to further highlight the issue of the flat Earth. Performance problem, p.e. physical preparation was also highlighted. With these comments, we are updating the play to seek the best show' • 'An importance in the dissemination of sciences through Art, not only opens a space for science, but also presents to an audience that is not frequent in any theater or theater.' what is not present in the theater is science, the audience is always present. • I suggest to rewrite more clearly: 'When you get stronger and do an artistic work based on the experiment to further guarantee the quality of scientific dissemination.' • I suggest to rewrite more clearly: 'Scientific dissemination should be one of the main functions for researchers, who normally participate as collaborators in most dissemination activities.' • The references should be more carefully organized conform the rules of the journal (https://www.geoscience-communication.net/for_authors/manuscript_preparation.html): doi numbering, comma and year. Also when referring to a webpage (example: Copernicus Publications: https://publications.copernicus.org/, last access: 25 October 2018.) • I was not able to access the link presented in the last reference as a 4shared document (Thomazoni W. 2013. A historia do universo – lyrics available (In Portuguese) http://www.4shared.com/mp3/Sv8mQnik . 2013). I suggested to use the one from youtube (https://www.youtube.com/watch?v=chvzmXTOBGs)

---

## Author Comment (AC1) · 17 Jun 2020

Dear Juliana Garrido Damaceno

We thank you for all your suggestions and the constructive comments for this paper. All of the issues that you pointed will be in our revision.

Sincerely,
* * *

---

## Referee Comment (RC2) · Anonymous Referee #2 · 18 Jun 2020

The paper "The Flat Earth satire, using science theatre to debunk absurd theories" deals with a way to answer to the flat Earth theory through a comic theatre show. The idea and scope of the show are certainly interesting and praiseworthy. However, the paper completely fails to give the show justice. First of all, the form is unsuitable for a scientific paper: the English language is plenty of mistakes, many sentences are too long or badly constructed, often unclear. Furthermore, the language is often more suitable to a newspaper than a scientific paper. I underlined some examples of sentences that need to be rewritten but actually the whole paper would need a professional scientific English editing. Another big issue is the content. The paper only superficially scratches what the show is about. There is no way to have an independent opinion about the effectiveness of the show, nor it is possible to reproduce it (which is

a fundamental part of a revision process). In this way the usefulness to the readers is also questionable. Finally, the references should also include some studies about science and art, communication and teaching. These are fields where researches are certainly not lacking.

Following are some specific comments. For future submissions please keep the line numbers throughout all the document: Line 16: you talk about theories but the fact that the Earth is not flat is more than a theory. Please correct. Line 26-28: correct the English as follows: "The show impacts the community in a funny way, offering the opportunity for a new experience to the population" 46-47: please rephrase and check English. 47: what do you mean with "Sciences have no actions or initiatives"? This seems like a slogan. Please, explain it or delete it. 49: I might disagree. There is plenty of books and scientific divulgators on TV addressing adults. My impression is quite the opposite. Please cite some data or sources backing up your statement or change it. 49-52: check the punctuation, the sentence is too long. 52-56: I believe you mean exhausting and not exhaustive. In any case it is a self-pitying consideration. This whole sentence should be rewritten in correct English and in a more appropriate language for a scientific journal. 66: cited as Palma in the reference list. 80: you must explain what is a mamulengo. 81: will be disturbed? Please explain better. And the slides? Explaining what? 92: white, not wine. 81-98: this is a very generical description of the show. In practice we have no idea of the contents and the readers cannot decide for themselves whether the show can be effective or not. Also, it is not possible to reproduce it. Page 4: what do you mean here "happened through several partners"? Page 4: this sentence is not clear (the audience leads you to a greater understanding?); moreover, the "understanding with laughter" is a slogan, not suitable for a scientific journal: "This result brings us to believe that we have an audience with excellent knowledge and that apparently can lead us to greater understanding with laughter". Page 4: here you introduce new elements, such as that the show is not ready, but it is not clear why. The whole paragraph should be rewritten: "As for the comments, the highlight is that the show is not ready yet, although all this result, especially the end

of the show and why we use of the "mamulengos". That it was necessary in order to highlight the training of the actor in terms of qualification in science. The need to further highlight the issue of the flat Earth. Performance problem, p.e. physical preparation was also highlighted. With these comments, we are updating the play to seek the best show." Page 5-6: again, this is not a language suitable for a scientific paper: "participate in these activities, there is no recognition or accounting for your academic careers".

---

## Referee Comment (RC3) · Aleksander Väljamäe (Referee) · 7 Jul 2020

The paper "The flat Earth satire..." touches an important topic of art-science interventions for fighting pseudoscientific beliefs. It has its potential but needs a major content revision, plus English language review. The main key points of improvement, if the paper will be re-submitted, are as follows: - It would be good to know about the previous art-science process in theatre and since the references are mainly in Portugese, I would encourage authors to extend the highlights from these earlier studies. - Introduction should be made stronger addressing the available statistics on Modern flat-Earthers etc. - I would recommend to create a new section that is dedicated to the show creation, with more details to understand what it was. On line 90 authors write that it was evaluated weakly - how, by who, and for what? Descrip-

tion of the presenter costume is detailed but how it is important? How the structure of the show was designed (e.g., mamulengos, different concepts) and whether there was some specific dramatically to it? - I would recommend a separate section on the questionnaire - why these particular questions were selected, what were the expectations/assumptions/hypotheses behind that? - Discussion needs some significant substance. Perhaps there were also some discussions with the audience, or semi-structured interviews. Any new performances planed? Recollections from the actors. Comments on the associated communication - posters, posts in social media. Again, possible reflections in press/social media? - It would be good if more reflections are provided on the collaborative process of the team (who was in the team) while creating the show. - Regarding the figures - I guess the original questions were in Brasilian. Please provide these as well, and check the translation - as it is now it is rather ambiguous. X-axis needs labels. Figure 3 is not really scientific - this percentage can be just mentioned in the text. Figure 4 - change the font, not easy to read now. Would be good to have the split between females and males (or age, education) for the statistics, if that was also collected. Would be nice to have more insights into Figure 6 - why these mixed responses - perhaps, if analyzed with the other data, it could be more informative (correlation between responses of previous knowledge or performance appreciation).

---

## Author Comment (AC2) · 25 Nov 2020

We appreciate all comments. I inform you that we are reviewing all English and all recommendations forwarded.

All responses will be in conjunction with all comments and suggestions sent by the Referres.

.

---

## Author Comment (AC3) · 26 Nov 2020

Referee 1: Specific comments: The work won the DF 2019 Theater Award Category: Local Circulation Show and this information should be present in the introduction
**Answer**: We insert in the introduction

Referee 1:The results and conclusion should be elaborated highlighting the fact that part of the public lacks basic knowledge of scientific information. . ..
**Answer**: We thanks, we improved the results and conclusion sessions

Referee 1: Please add more information about the public that filled the form (age, level of education, nationality, etc)
**Answer**: Unfortunately we didn't insert this question in our form

[Figure]

Referee 1: I appreciated the conclusions but I also expected to know more about the next steps in developing this project.
**Answer**: We improved the conclusion.

Referee 1: I suggest to remove the phrase 'Sciences have no actions or initiatives
**Answer**: We agreed and removed

Referee 1: I suggest to rewrite more clearly the paragraph that starts in line 52: 'Searching for this connection between science and art is certainly complicated, due to the exhaustive scientific activity of researchers, which most of the time there is no way to present themselves in the art form, however, we can unite with educating artists and assist or boost our art science.
**Answer**: We rewrote this sentence

Referee 1: I suggest to rewrite more clearly the paragraph that starts in line 59: ' . That seeks this link through the clowning between science and art and in this work the whole process of creation will be shown in a summarized form and also the results obtained from the audience through the google form for the public that attended.'
**Answer**: We rewrote this sentence

Referee 1: I suggest to rewrite more clearly the paragraph that starts in line 74: 'Our work enters the area of earth science and art, where it is still very little explored and we use the same tools used by previous works (what tools?) The information was collected on this subject using the social networks that defended this false theory and through scientific dissemination that refute the theme, scenes for spectacle were simultaneously created.'
**Answer**:We rewrote this sentence

Referee 1: I suggest to rewrite more clearly the paragraph that starts in line 79: '. The show's introduction scenes were selected, in which we used two "mamulengos" to present the show.' Please could you explain what 'mamulengos' are?
**Answer**:We inserted what "mamulengos" are

Referee 1: I suggest to rewrite more clearly the paragraph that starts in line 81: 'After the presentation, a short lecture that will be disturbed and at least three slides. For the next scene, we break the text of the big bang theory, adapted from Oliveira 2018, in which we use the body as language. Given continuity, music inspired by the opening of the series The Big Bang Theory, by Thomazoni W. (2013) was used and adapted for the show. In addition, scenes from França et al. 2019 (Figure 1) featured Newton's scene and gravity (please we need more explanation about) in the EGU session. Finally, the remaining three (or two?) scenes were: the story of the planets with balloons and a Pilates ball; based on the film "the great dictator" by Charlie Chaplin; and finally the Earth is Flat sermon'
**Answer**: We rewrote this sentence

Referee 1: I suggest to rewrite more clearly the paragraph that starts in line 96: 'With everything ready, leaving only the scenario we chose in a scenario that facilitated mobility'
**Answer**:We rewrote this sentence

Referee 1: The reference lines disappeared on page 4
**Answer**: Sorry about this, we will insert correctly the reference lines for the final version

Referee 1: I suggest to rewrite more clearly: 'we used the audience that watched through a google form that was made available. This form had 9 questions and an optional comment Of the 316 people who have watched the show so far, 11**Answer**: We rewrote this sentence

Referee 1: I suggest to rewrite more clearly: 'This division shows that we had a well-distributed response and there was a mix of the audience from the academy and also an audience, since the presentation in Natal'
**Answer**: We rewrote this sentence

Referee 1: Add parentheses (Figure 3a) in: 'certainly the title makes the public more aware of the subject Figure 3a.

**Answer**: We added, thank you for this warning

Referee 1: I suggest to rewrite more clearly: 'This result brings us to believe that we have an audience with excellent knowledge and that apparently can lead us to a greater understanding with laughter.'
**Answer**:We rewrote this sentence

Referee 1: Authors should be more careful with punctuation and double spacing;
**Answer**: Ok, We reviewed our punctuations and double spacing

Referee 1: I suggest to rewrite more clearly: 'The second part is about on scientific dissemination, the first question is one word was asked to represent the show, figure 4 shows a word cloud with emphasis on didactic, interesting, fun, playful, genius. This shows that we had a general acceptance of the public. If the show is science dissemination with a score of 1 to 5, 83**Answer**: We rewrote this sentence

Referee 1: I suggest to rewrite more clearly: 'The show regarding classification and recommendation to another person obtained results similar to that of scientific dissemination, with another 80**Answer**: We rewrote this sentence

Referee 1: I suggest to rewrite more clearly: 'As for the comments, the highlight is that the show is not ready yet, although all this result, especially the end of the show and why we use the "mamulengos". That it was necessary in order to highlight the training of the actor in terms of qualification in science. The need to further highlight the issue of the flat Earth. Performance problem, p.e. physical preparation was also highlighted. With these comments, we are updating the play to seek the best show'
**Answer**: We rewrote this sentence

Referee 1: An importance in the dissemination of sciences through Art, not only opens a space for science but also presents to an audience that is not frequent in any theater or theater.' what is not present in the theater is science, the audience is always present
**Answer**:We rewrote this sentence and tried to explain the kind of audience

Referee 1: I suggest to rewrite more clearly: 'When you get stronger and do an artistic work based on the experiment to further guarantee the quality of scientific dissemination.
**Answer**: We rewrote this sentence

Referee 1: I suggest to rewrite more clearly: 'Scientific dissemination should be one of the main functions for researchers, who normally participate as collaborators in most dissemination activities
**Answer**: We rewrote this sentence

Referee 1: The references should be more carefully organized confirm the rules of the journal (https://www.geoscience−communication.net/forauthors/manuscriptpreparation.html): doi numbering, comma and year. Also when referring to a webpage (example: Copernicus Publications: https://publications.copernicus.org/, last access: 25 October 2018.)
**Answer**: We fixed all references.

Referee 1: I was not able to access the link presented in the last reference as a 4shared document (Thomazoni W. 2013. A história do universo – lyrics available (In Portuguese) http://www.4shared.com/mp3/Sv8mQnik . 2013). I suggested to use the one from youtube (https://www.youtube.com/watch?v=chvzmXTOBGs)
**Answer**: We agreed and used your suggestions!

---

## Author Comment (AC4) · 26 Nov 2020

Dear Referee

Below our point-to-point response and comments

First of all, the form is unsuitable for a scientific paper: the English language is plenty of mistakes, many sentences are too long or badly constructed, often unclear. Furthermore, the language is often more suitable to a newspaper than a scientific paper. I underlined some examples of sentences that need to be rewritten but actually, the whole paper would need a professional scientific English editing.

[Figure]

**Answer :** We reviewed professional Scientific English.

The paper only superficially scratches what the show is about. There is no way to have an independent opinion about the effectiveness of the show, nor it is possible to reproduce it (which is Printer-friendly version Discussion paper a fundamental part of a revision process). In this way the usefulness to the readers is also questionable. Finally, the references should also include some studies about science and art, communication and teaching. These are fields where researchers are certainly not lacking.
**Answer :** We reviewed references and we inserted the art reviews

Line 16: you talk about theories but the fact that the Earth is not flat is more than a theory. Please correct.
**Answer :** we fixed this

Line 26-28: correct the English as follows: "The show impacts the community in a funny way, offering the opportunity for a new experience to the population"
**Answer :** "The show impacts the audience in a fun way, offering the opportunity for a new experience for the population"

46-47: please rephrase and check English
**Answer :**We rephrase this sentence and checked English

47: what do you mean with "Sciences have no actions or initiatives"? This seems like a slogan. Please, explain it or delete it.
**Answer :** We deleted it

49: I might disagree. There is plenty of books and scientific divulgators on TV addressing adults. My impression is quite the opposite. Please cite some data or sources backing up your statement or change it.
**Answer :**This sentence is special for Brazil or made by Brazilians, but I found more citations and inserted .

49-52: check the punctuation, the sentence is too long.

**Answer :** we rewrote

52-56: I believe you mean exhausting and not exhaustive. In any case it is a self-pitying consideration. This whole sentence should be rewritten correctly in English and in a more appropriate language for a scientific journal.
**Answer :** we rewrote

66: cited as Palma in the reference list.
**Answer :** We fixed

80: you must explain what mamulengo is.
**Answer :** we explained and improved this sentence

81: will be disturbed? Please explain better. And the slides? Explaining what?
**Answer :** We rewrote this sentence

92: white, not wine.
**Answer :** It is wine color

81-98: this is a very generical description of the show. In practice we have no idea of the contents and the readers cannot decide for themselves whether the show can be effective or not. Also, it is not possible to reproduce it.
**Answer :** We rewrote with more details

Page 4: what do you mean here "happened through several partners"?
**Answer :** We rewrote this sentence

Page 4: this sentence is not clear (the audience leads you to a greater understanding?); moreover, the "understanding with laughter" is a slogan, not suitable for a scientific journal: "This result brings us to believe that we have an audience with excellent knowledge and that apparently can lead us to greater understanding with laughter".
**Answer :** We rewrote this sentence

Page 4: here you introduce new elements, such as that the show is not ready, but it

is not clear why. The whole paragraph should be rewritten: "As for the comments, the highlight is that the show is not ready yet, although all this result, especially the end of the show and why we use of the "mamulengos". That it was necessary in order to highlight the training of the actor in terms of qualification in science. The need to further highlight the issue of the flat Earth. Performance problem, p.e. physical preparation was also highlighted. With these comments, we are updating the play to seek the best show."

**Answer :** We rewrote this phrase

Page 5-6: again, this is not a language suitable for a scientific paper: "participate in these activities, there is no recognition or accounting for your academic careers".
**Answer :** We remove this sentence

---

## Author Comment (AC5) · 26 Nov 2020

Dear Referee

Below our point-to-point response and comments

It would be good to know about the previous art-science process in theatre and since the references are mainly in Portuguese, I would encourage authors to extend the highlights from these earlier studies.

**Answer:** This is our first experience and we reviewer English

- Introduction should be made stronger addressing the available statistics on Modern

flat-Earthers etc.

**Answer:** We didn't find any statistics about flat-earthers only in social media or superficial news. we are still research this topic. Thank you for this issue

- I would recommend creating a new section that is dedicated to the show creation, with more details to understand what it was.

**Answer:** We rewrote but inserted in the Art-Science and the process section.

On line 90 authors write that it was evaluated weakly - how, by who, and for what?

**Answer:** we inserted with weekly rehearsals

Description of the presenter costume is detailed but how it is important?

**Answer:**We inserted

How the structure of the show was designed (e.g., mamulengos, different concepts) and whether there was some specific dramatically to it?

**Answer:**We rewrote this part

- I would recommend a separate section on the questionnaire - why these particular questions were selected, what were the expectations/assumptions/hypotheses behind that?

**Answer:**We agreed with you, but we didn't do this yet.

- Discussion needs some significant substance. Perhaps there were also some discussions with the audience, or semi-structured interviews. Any new performances planed? Recollections from the actors.

**Answer:** Ok, We rewrote the discussion

Comments on the associated communication - posters, posts in social media.

Again, possible reflections in press/social media? - It would be good if more reflections are provided on the collaborative process of the team (who was in the team) while creating the show.

- Regarding the figures - I guess the original questions were in Brasilian. Please provide these as well, and check the translation - as it is now it is rather ambiguous. X-axis needs labels.
**Answer:** We inserted

Figure 3 is not really scientific - this percentage can be just mentioned in the text.
**Answer:**OK — I removed this picture.

Figure 4 - change the font, not easy to read now. Would be good to have the split between females and males (or age, education) for the statistics,
**Answer:** Thanks, the issue is possible in the next opportunity because in this questionnaire didn't have this questions about age, education...

if that was also collected. Would be nice to have more insights into

Figure 6 - why these mixed responses - perhaps, if analyzed with the other data, it could be more informative (correlation between responses of previous knowledge or performance appreciation).
**Answer:** We wrote about Figure 5? Word cloud! Did you write about Figure 5? Word cloud! This is the only an illustration for they were feeling about the show.

---

## Author Response (AR1)

Dear Editor Dr. Francesco Mugnai,

Many thanks for sending us the comments from the referees. We appreciate the suggestions for improving our manuscript and have tried to address them as best as we could.

We are submitting the revised manuscript. Below, you can find your and the referees' comments in black, and our responses in blue. Major revisions and added text in the manuscript are indicated in yellow and number[ ] . We look forward to hearing from you whether additional revisions are necessary.

Sincerely,
(on behalf of all authors).
* * *
**For all referees**

We thank all the referees for pointing us to important issues in the submitted manuscript. Following your suggestions along with those from Referee 1 to Referee 3, we have carefully addressed all outstanding issues. As a result, we believe that as it is, the new manuscript version is in better shape.

**Referee 1: Juliana Garrido Damaceno (Referee)**

Comments to the Author(s)

Specific comments: The work won the DF 2019 Theater Award Category: Local Circulation Show and this information should be present in the introduction; The results and conclusion should be elaborated highlighting the fact that part of the public lacks basic knowledge of scientific information….--> Thanks, we inserted the award in the introduction, improved the results and conclusion sessions.[1]

Please add more information about the public that filled the form (age, level of education, nationality, etc) --> Unfortunately we did not ask these questions in the form.

I appreciated the conclusions but I also expected to know more about the next steps in developing this project. --> We wrote a new paragraph at the end of the conclusion section.[2]

I suggest to remove the phrase 'Sciences have no actions or initiatives' --> We agreed and removed it. [3]

I suggest to rewrite more clearly the paragraph that starts in line 52: 'Searching for this connection between science and art is certainly complicated, due to the exhaustive scientific activity of researchers, which most of the time there is no way to present themselves in the art form, however we can unite with educating artists and assist or boost our art science.'[4]

I suggest to rewrite more clearly the paragraph that starts in line 59: ' . That seeks this link through the clowning between science and art and in this work the whole process of creation will be shown in a summarized form and also the results obtained from the audience through the google form for the public that attended.' [5]

I suggest to rewrite more clearly the paragraph that starts in line 74: 'Our work enters the area of earth science and art, where it is still very little explored and we use the same tools used by previous works (what tools?) The information was collected on this subject using the social networks that defended this false theory and through scientific dissemination that refute the theme, scenes for spectacle were simultaneously created.' --> We rewrote all these sentences[6]

I suggest to rewrite more clearly the paragraph that starts in line 79: '. The show's introduction scenes were selected, in which we used two "mamulengos" to present the show.' Please could you explain what 'mamulengos' are? ---> We inserted what "mamulengos" are in line 45-49 [7]

I suggest to rewrite more clearly the paragraph that starts in line 81: 'After the presentation, a short lecture that will be disturbed and at least three slides. For the next scene, we break the text of the big bang theory, adapted from Oliveira 2018, in which we use the body as language. Given continuity, music inspired by the opening of the series The Big Bang Theory, by Thomazoni W. (2013) was used and adapted for the show. In addition, scenes from França et al. 2019 (Figure 1) featured Newton's scene and gravity (please we need more explanation about) in the EGU session. Finally, the remaining three (or two?) scenes were: the story of the planets with balloons and a Pilates ball; based on the film "the great dictator" by Charlie Chaplin; and finally the Earth is Flat sermon' [8]

I suggest to rewrite more clearly the paragraph that starts in line 96: 'With everything ready, leaving only the scenario we chose in a scenario that facilitated mobility' --> We rewrote all these sentences [9]

The reference lines disappeared in page 4 --> Sorry about this, we will insert the reference lines correctly in the final version

I suggest to rewrite more clearly: 'we used the audience that watched through a google form that was made available. This form had 9 questions and an optional comment Of the 316 people who have watched the show so far, 11% of the public have answered the questionnaire'[10]

I suggest to rewrite more clearly: 'This division shows that we had a well distributed response and there was a mix of the audience from the academy and also an audience, since the presentation in Natal' --> We rewrote all these sentences[11]

Add parentheses (Figure 3a) in: 'certainly the title makes the public more aware of the subject Figure 3a. --> We added, thank you for this warning

I suggest to rewrite more clearly: 'This result brings us to believe that we have an audience with excellent knowledge and that apparently can lead us to greater understanding with laughter.' --> We rewrote this sentence [12]

Authors should be more careful with punctuation and double spacing; --> Ok, We checked our punctuations and double spacing

I suggest to rewrite more clearly: 'The second part is about scientific dissemination, first question is one word was asked to represent the show, figure 4 shows a word cloud with emphasis on didactic, interesting, fun, playful, genius. This shows that we had a general acceptance of the public. If the show is science dissemination with a score of 1 to 5, 83% gave a score of 4 and 5 and 17% gave a score of 3, again it shows the importance of this play with a link in a space that is at least used in Federal Capital in Brazil'[13]

       I suggest to rewrite more clearly: 'The show regarding classification and recommendation to another person obtained results similar to that of scientific dissemination, with another 80% with a score between 4 and 5. What we can conclude was that it was well accepted by the public (Figure 7).' [14]

       I suggest to rewrite more clearly: 'As for the comments, the highlight is that the show is not ready yet, although all this result, especially the end of the show and why we use the "mamulengos". That it was necessary in order to highlight the training of the actor in terms of qualification in science. The need to further highlight the issue of the flat Earth. Performance problem, p.e. physical preparation was also highlighted. With these comments, we are updating the play to seek the best show' --> We rewrote all these sentence [15]

       An importance in the dissemination of sciences through Art, not only opens a space for science, but also presents to an audience that is not frequent in any theater or theater.' what is not present in the theater is science, the audience is always present --> We rewrote this sentence and explained what type of audience watched the performances. [16]

I suggest to rewrite more clearly: 'When you get stronger and do an artistic work based on the experiment to further guarantee  the quality of scientific dissemination. [17]

I suggest to rewrite more clearly: 'Scientific dissemination should be one of the main functions for researchers, who normally participate as collaborators in most dissemination activities

--> We rewrote all these sentences [18]

The references should be more carefully organized conform the rules of the journal (https://www.geoscience-communication.net/for_authors/manuscript_preparation.html): doi numbering, comma and year. Also when referring to a webpage (example: Copernicus Publications: https://publications.copernicus.org/, last access: 25 October 2018.) --> We fixed all references.

I was not able to access the link presented in the last reference as a 4shared document (Thomazoni W. 2013. A história do universo – lyrics available (In Portuguese) http://www.4shared.com/mp3/Sv8mQnik . 2013). I suggested to use the one from youtube  (https://www.youtube.com/watch?v=chvzmXTOBGs)  --> We agreed and used your suggestions!

**Referee 2: Anonymous**

First of all, the form is unsuitable for a scientific paper: the English language is plenty of mistakes, many sentences are too long or badly constructed, often unclear. Furthermore, the language is often more suitable to a newspaper than a scientific paper. I underlined some examples of sentences that need to be rewritten but actually the whole paper would need a professional scientific English editing. --> : We sent the paper to be revised by a Scientific English professional .

The paper only superficially scratches what the show is about. There is no way to have an independent opinion about the effectiveness of the show, nor it is possible to reproduce it (which is Printer-friendly version Discussion paper a fundamental part of a revision process). In this way the usefulness to the readers is also questionable. Finally, the references should also include some studies about science and art, communication and teaching. These are fields where researchers are certainly not lacking. --> We reviewed the references and inserted the art reviews

Line 16: you talk about theories but the fact that the Earth is not flat is more than a theory. Please correct. --> Thank you and we fixed this

Line 26-28: correct the English as follows: "The show impacts the community in a funny way, offering the opportunity for a new experience to the population" --> "The show impacts the audience in a fun way, offering the opportunity for new experience for the population"[19]

46-47: please rephrase and check English --> We rephrase this sentence and checked English [20]

47: what do you mean with "Sciences have no actions or initiatives"? This seems like a slogan. Please, explain it or delete it. --> We deleted it [3]

49: I might disagree. There is plenty of books and scientific divulgation on TV addressing adults. My impression is quite the opposite. Please cite some data or sources backing up your statement or change it. --> This sentence is special for Brazil or made by Brazilians, but I found more citations and inserts.

49-52: check the punctuation, the sentence is too long.
52-56: I believe you mean exhausting and not exhaustive. In any case it is a self-pitying consideration. This whole sentence should be rewritten in correct English and in a more appropriate language for a scientific journal. --> --> we rewrote the whole sentence [4]

66: cited as Palma in the reference list. --> Ok

80: you must explain what mamulengo is. --> we explained and improved this sentence [7]

81: will be disturbed? Please explain better. And the slides? Explaining what?
- > We rewrote this sentence [8]

92: white, not wine. --> I use wine-colored .[21]

81-98: this is a very generical description of the show. In practice we have no idea of the contents and the readers cannot decide for themselves whether the show can be effective or not. Also, it is not possible to reproduce it. --> We rewrote with more details[22]

Page 4: what do you mean here "happened through several partners"? --> We rewrote this sentence [23]

Page 4: this sentence is not clear (the audience leads you to a greater understanding?); moreover, the "understanding with laughter" is a slogan, not suitable for a scientific journal: "This result brings us to believe that we have an audience with excellent knowledge and that apparently can lead us to greater understanding with laughter". --> We rewrote this sentence [12]

Page 4: here you introduce new elements, such as that the show is not ready, but it is not clear why. The whole paragraph should be rewritten: "As for the comments, the highlight is that the show is not ready yet, although all this result, especially the end of the show and why we use of the "mamulengos". That it was necessary in order to highlight the training of the actor in terms of qualification in science. The need to further highlight the issue of the flat Earth. Performance problem, p.e. physical preparation was also highlighted. With these comments, we are updating the play to seek the best show." --> We rewrote this phrase [15]

Page 5-6: again, this is not a language suitable for a scientific paper: "participate in these activities, there is no recognition or accounting for your academic careers". --> We removed this sentence

**Referee 3 : Aleksander Väljamäe**

It would be good to know about the previous art-science process in theatre and since the references are mainly in Portuguese, I would encourage authors to extend the highlights from these earlier studies. --> Thanks, here we explain our experience during this work and how the process works, in the next opportunity we will review the art-science process.

- Introduction should be made stronger addressing the available statistics on Modern flat-Earthers etc. --> We didn't find any statistics on flat-earthers only in the social media or superficial news. We are still researching this topic. Thank you for raising this point.

- I would recommend to create a new section that is dedicated to the show creation, with more details to understand what it was. --> We rewrote and inserted in the Art-Science and the process section.

On line 90 authors write that it was evaluated weakly - how, by who, and for what? --> we inserted with weekly rehearsals [8]

Description of the presenter costume is detailed but how it is important? --> We inserted [23]

How the structure of the show was designed (e.g., mamulengos, different concepts) and whether there was some specific dramatically to it? --> We rewrote this part [8]

- I would recommend a separate section on the questionnaire - why these particular questions were selected, what were the expectations/assumptions/hypotheses behind that? --> We agreed with you, in the next opportunity we will do

- Discussion needs some significant substance. Perhaps there were also some discussions with the audience, or semi-structured interviews. Any new performances planed? Recollections from the actors. --> Ok, We rewrote the discussion

Comments on the associated communication - posters, posts in social media. Again, possible reflections in press/social media? - It would be good if more reflections are provided on the collaborative process of the team (who was in the team) while creating the show.

--> All authors are part of the team and we inserted more details about the process in the paper.

- Regarding the figures
- I guess the original questions were in Brasilian. Please provide these as well, and check the translation - as it is now it is rather ambiguous.

X-axis needs labels.

--> We inserted

Figure 3 is not really scientific - this percentage can be just mentioned in the text.

--> OK --- I removed this picture

Figure 4 - change the font, not easy to read now. Would be good to have the split between females and males (or age, education) for the statistics,

--> Thanks, the issue is possible in the next opportunity because our questionnaire didn't have these questions.

if that was also collected. Would be nice to have more insights into

Figure 6 - why these mixed responses - perhaps, if analyzed with the other data, it could be more informative (correlation between responses of previous knowledge or performance appreciation).

--> Now, it is Figure 5. This is only an illustration to demonstrate how they were feeling about the show.

---

## Author Response (AR2)

Dear Editor Dr. Francesco Mugnai,

We are grateful to the editor for considering our paper for publication. We corrected the figures and inserted the number for each figure. Also, we used the same WORD template that it available in https://www.geoscience-communication.net/submission.html#templates and follow the recommendations.

Your Sincerely,
(on behalf of all authors).